# Household Resilience to Food and Nutrition Insecurity in Central America and the Caribbean

Ricardo Sibrian [1], Marco d'Errico [2,*], Patricia Palma de Fulladolsa [1] and Flavia Benedetti-Michelangeli [2]

[1] Programme on Information Systems for Food and Nutrition Security Resilience in the SICA Region Phase II, General Secretariat of the Central American Integrating System (PROGRESAN-SICA II), Antiguo Cuscatlán 01-113, El Salvador; rsibrian@sica.int (R.S.); ppalma@sica.int (P.P.d.F.)

[2] Agrifood Economics Division, FAO, 00153 Rome, Italy; fla.benedettimich@stud.uniroma3.it

[*] Correspondence: marco.derrico@fao.org

**Abstract:** Latin American and Caribbean countries, affected mainly by extreme climatic events, are heterogeneous in farming practices and the relevance of critical determinants of resilience. This paper fills the knowledge gap and informs on the application of the Resilience Index Measurement and Analysis version II (RIMA-II) for estimating Resilience on Food and Nutrition Security (RFNS) indicators in five vulnerable countries in Central America and the Caribbean: Costa Rica, El Salvador, Guatemala, Honduras, and the Dominican Republic. Already-collected information on food consumption and social and economic dimensions, depicting key determinants or "pillars" as defined by RIMA-II methodology, is the basis for developing several models on RFNS. These findings are baselines for subnational territories and country-specific inputs for monitoring and enhancing Food and Nutrition Security (FNS) indicators. This paper fills three critical gaps in the literature on resilience. It presents cross-country data-driven evidence, highlighting consistencies and discrepancies by analyzing data on otherwise unexplored Latin American and Caribbean countries. It suggests the country-specific approach of resilience measurement for heterogeneous contexts. In addition, it provides policy indications to support the role of farm diversification in promoting household resilience.

**Keywords:** resilience; food and nutrition security; structural equation models





## 1. Introduction

Natural, economic, and political factors, including corruption and diversion of resources from public policies, reduce food and nutrition security. Consequently, resilience is lower in rural and urban areas [1,2].

The term 'resilience' has been widely adopted within policies, programming, and thinking around climate change adaptation ('adaptation') and disaster risk reduction—DRR [3]. As interest and financial investments in resilience have grown, many measurement approaches emerged that created a vast data-driven literature. Findings from these analyses provide decision-makers with inputs for targeting populations and prioritizing actions for supporting public policies toward improving the food and nutrition security of vulnerable people.

This paper uses the Resilience Index Measurement and Analysis (RIMA) outlined in FAO 2016 [4] to estimate Resilience on Food and Nutrition Security (RFNS) indicators. It builds upon the definitions and approaches to measure resilience as endorsed by the Technical Working Groups on Resilience Measurement and drafted in Constas and Barrett [5]. In particular, RIMA is designed around the following definition of resilience: the capacity that ensures shocks and stressors do not have long-term developmental consequences.

This paper contributes to the literature on resilience measurement in three ways. It fills the gap of (1) cross-countries analyses (2) for an unexplored area (Latin America and the Caribbean) and (3) provides evidence on the role of farm diversity in building resilience.

The increasing interest in resilience analysis has recently boosted the necessity to develop affordable estimation methods to produce a cross-country comparison [6]. Significant emphasis has been placed on finding culturally transferable measures of resilience that provide valid comparisons across contexts [7–9] and identifying a set of attributes, processes, that hold across all conditions along with those that depend on local conditions [6]. The nature and relative importance of objective indicators for resilience capacities vary between shock/stressor types, livelihood contexts, and cultures [9–11]. There is scarce data-driven cross-countries evidence on context-specific resilience drivers. d'Errico, Pietrelli, and Romano in 2016 [12] is the only work on household resilience in more than one country (Uganda and Tanzania). Generally, policies reflecting local conditions are crucial in addressing food insecurity [13] and country-specificities, particularly relevant policy design and resilience-enhancing interventions.

Extreme climate change in Central America and the Caribbean impacts food production such as maize, beans, rice, and sorghum as well income-generating activities [14,15]. The impact is so strong that neither self-food consumption for smallholder farmers in rural areas nor food purchase for labor in rural areas and marginal urban communities are appropriate for food and nutrition security, particularly in vulnerable populations [16]. Moreover, Latin American and Caribbean countries are highly heterogeneous in agriculture and vulnerability both among and within countries [17].

This paper provides evidence for another vital research question: the possible relevant role of farm diversity in resilience-enhancing efforts. Farming is still the main economic activity in rural areas and the primary source of employment for the economically active population in these regions. There is ample evidence of the links between farm diversity and resource use efficiency. Variety assumes here two meanings: a heterogeneous approach to farming practice, one of agriculture's principal assets to respond to uncertain futures, and alternatively, context-specific response mechanisms. The challenges for agroecological transitions are not the same for all farmers in every area or context; they can face different social and biophysical conditions.

Farm diversity is positively correlated with household diet diversity and nutritional quality, especially for ultra-poor, subsistence-oriented farmers. Diversification is a primary livelihood strategy for improving food and nutrition security, given the pervasiveness of agriculture and farm in rural areas [18]. The combined effect of resilience-enhancing initiatives with agroecological interventions can effectively reduce the adverse effects of shocks on wellbeing, thus being associated with higher recovery capacity [19]. Policies addressed to small farms in rural areas need to support diversified farming systems to specific contexts [20]. Indeed, resilience is associated with diversification systems worldwide and even within the same region [20].

The remainder of the paper is organized as follows. The next chapter introduces the data and methods employed. Then, results are shown and discussed. Finally, the conclusion and policy implications are presented.

## 2. Data and Methods

### 2.1. Data

Survey data from the National Household Income and Expenditure Survey (Costa Rica, 2013), National Household Multi-Purposes Survey (El Salvador, 2015), National Household Living Standard Measurement Survey (Guatemala, 2014), National Household Living Standard Measurement Survey (Honduras, 2004), and National Household Income and Expenditure Survey (the Dominican Republic, 2005–2006) were the bases for the analyses. Table 1 provides features of these surveys in terms of institutions in charge of data collection, year, type of survey, sample size, coverage, date of data collection, and questionnaire content. All surveys represent national, urban–rural, and subnational levels (territories in El Salvador, Guatemala, and Honduras, Costa Rica, and macro-regions in the Dominican Republic).



**Table 1.** Features of household surveys.

| Household Survey Features | Costa Rica | Dominican Republic | El Salvador | Guatemala | Honduras |
|---|---|---|---|---|---|
| Data collection institution | National Institute of Statistics and Census (INEC) | National Statistics Office (ONE) | Department of Statistics and Census (DIGESTYC) | National Statistics Institute (INE) | National Statistics Institute (INE) |
| Year | 2013 | 2005–2006 | 2015 | 2014 | 2004 |
| Type | Income and expenditure | Income and expenditure | Multi-purpose | Living standard measurement | Living standard measurement |
| Sample | 5623 out of 5627 (99.9%) | 8222 out of 9600 (85.6%) | 23,669 out of 23,670 (100.0%) | 11,317 out of 11,536 (98.1%) | 8121 out of 8175 (99.4%) |
| Coverage | National Urban and Rural areas (NUR) and regions | National Urban and Rural areas (NUR) and macro-regions (with 31 provinces and National District) | National Urban and Rural areas (NUR) and territories | National Urban and Rural areas (NUR) and territories | National Urban and Rural areas (NUR) and territories |
| Date | January–December spread monthly | January–December spread monthly | January–December spread monthly | January–December spread monthly | July–November spread monthly |
| Questionnaire | Demography, income and income sources, the expenditure of goods and services, and other information related to social safety nets | Demography, housing, education, health, employment, agricultural production, income and income sources, the expenditure of goods and services, and other information related to social safety nets | Demography, education, ICTs, housing, employment and income, agricultural production, health, family transactions, the expenditure of goods and services, subsidies, childhood and youth conditions | Demography, the cost of goods and services, education, health, migration, labor market, housing, household food insecurity experience, childhood and adolescence conditions | Demography, income, and expenditure of goods and services, education, health, migration, labor market, housing |

Source: Authors' own elaboration.

Specifically, as shown in Table 1, National Institutes of Statistics (INE) provides data for Honduras and Guatemala, while the National Institute of Statistics and Census (INEC), Department of Statistics and Census (DIGESTYC), and National Statistics Office (ONE) provide for Costa Rica, El Salvador, and the Dominican Republic, correspondingly. The response rate in El Salvador is 100%, followed by Costa Rica (99.9%) and Honduras (99.4%). The lowest response rate is Guatemala with 98.1% and the Dominican Republic with 85.6%. The data collection period is from January to December (spread monthly) for all countries, except Honduras (survey ends in November).

Generally, all questionnaires include demographics, expenditures for goods and services, and income sources sections. In addition, housing, education, labor market, employment, and health sections are relevant across countries, except Costa Rica. For consistency to country specificity, the survey content can differ across countries. As Table 1 shows, the migration section is developed in Guatemala and Honduras surveys, while agricultural production in the Dominican Republic and El Salvador. Information on Social Safety Nets is reported in Costa Rica and the Dominican Republic.

*2.2. Methods*

This paper makes use of one of the most widely accepted tools for measuring development resilience [21,22]. The initial model proposed for assessing resilience [23] included

household capacity for sustaining wellbeing, away from the effects of shocks and stresses, in general. In 2013, the Technical Working Group on Resilience Measurement defined resilience in similar terms as the household capacity to tackle stressors and shocks so that no long-lasting adverse development consequences occur [6]. FAO proposed improvements to the original resilience model by having food and nutrition security indicators for reflecting resilience depicted by a Multiple Effects Indicators Multiple Causes (MIMIC) model [4]. Resilience expressed with poverty indicators refers to the resilience of financial and economic wellbeing, food, and nutrition insecurity indicators that refer to the resilience of food and nutrition. In this sense, resilience emphasizes the household capacity to absorb, adapt, and transform coping strategies against the effects of shocks and stresses on food and nutrition insecurity.

Resilience measurement and analysis in the context of food security have witnessed a clear evolution: RIMA-II is the first tool to define resilience as a capacity, separated from the concept of food security [4]. To our knowledge, there are no innovative methods following the RIMA-II tool, if not for Barrett and Cissé's conditional moment-based approach for estimating household-level development resilience from panel data in 2018 [24]. Since 2016, several attempts have been made to review resilience. Serfilippi and Ramnath in 2017 [25] listed indicators as a reference point to enhance resilience understanding. Studies are mainly empirical and based on cross-sectional data, while only a few are empirical and analytical. In general, multivariate techniques dominate for quantifying resilience, but few regression-based approaches have also been developed [26]. Knippenberg, Jensen, and Constas in 2019 calculate resilience measures based on different approaches and compare the results through high-frequency data to examine the effects of household characteristics and shocks on food insecurity and welfare [27]. Notwithstanding, data-driven evidence is still needed for context-specificity in resilience measurement, and the RIMA-II model provides evidence for filling the gap.

The adopted FAO's RIMA-II model [4] measures resilience to food and nutrition insecurity reducing information of observed indicators into indices or key determinants for continuous indicators using confirmatory factor analysis (CFA) and for dichotomous indicators using polychoric confirmatory factor analysis (PCFA), capturing at least 95 percent variability of these substantive correlated and observed indicators.

A MIMIC structural regression model uses, as causes, unobserved key determinants of resilience: access to basic services (ABS), assets (AST), social safety nets (SSN), and adaptive capacity (AC) and, as effects, food and nutrition security (FNSI) indicators.

Figure A1 in the Appendix A depicts a RIMA-II model for an example with two FNSI. The two components of the MIMIC model, the measurement model with observed FNSI and the structural model with four (ABS, AST, SSN, and AC) key determinants of the resilience capacity index (RCI).

The equation for the structural model for all countries,

$$\text{RCI} = \text{ABS} \times \beta_{\text{abs}} + \text{AST} \times \beta_{\text{ast}} + \text{SSN} \times \beta_{\text{ssn}} + \text{AC} \times \beta_{\text{ac}} + \varepsilon_{\text{RCI}} \tag{1}$$

Causal coefficients $\beta_{\text{abs}}$, $\beta_{\text{ast}}$, $\beta_{\text{ssn}}$, and $\beta_{\text{ac}}$ are four critical determinants for resilience to food and nutrition insecurity, and disturbance of RCI is $\varepsilon_{\text{RCI}}$.

While the two food security indicators are modelled in the measurement part of the SEM as follows:

$$\text{FNS indicator 1} = \text{RCI} \times \Lambda_{\text{FNS1}} + \varepsilon_{\text{FNS1}} \tag{2}$$

$$\text{FNS indicator 2} = \text{RCI} \times \Lambda_{\text{FNS2}} + \varepsilon_{\text{FNS2}} \tag{3}$$

The $\Lambda_{\text{FNS1}}$ coefficient is scaled by setting it to 1 so that one standard deviation increase in RCI implies an increase of one standard deviation in FNS1. The unit of measure of coefficients for other FNS indicators (coefficient $\Lambda_{\text{FNS2}}$ and variance of the FNS) is determined by the scaling and is zero for var(FNS1) and var(FNS2). Measurement errors of FNS1 and FNS2 are $\varepsilon_{\text{FNS1}}$ and $\varepsilon_{\text{FNS2}}$, respectively.

Figure A1 in the Appendix A also shows observed indicators I(1), . . . , I(k) expressing unobserved ABS, AST, SSN, and AC, altogether with measurement errors e(1), . . . , e(k). These key determinants express at least 95 percent of the variation of their corresponding indicators, reduced in re-scaled indices, or re-scaled key determinants of resilience to food and nutrition insecurity.

## 3. Results and Discussion

### 3.1. Description of Results

Table 2 portrays achieved MIMIC models based on estimated key determinants derived from their corresponding observed indicators for participating countries. All models developed are different: the Costa Rican model has the primary vital determinants only (Figure A2 in the Appendix A), and the other four countries show interactions between critical determinants. The Guatemalan and Salvadorian models in Figure A2, and the Dominican model in Figure A3 in the Appendix A, include interaction between ASB and AST; the Honduran model in Figure A3 includes interactions between ABS and AST, AST and AC, and AC and SSN.

Observed FNS indicators available in national household surveys differ among countries. Models in El Salvador and the Dominican Republic (Figures A2 and A3, respectively) have two etic hands from a technical perspective, Costa Rica and Honduras (Figures A2 and A3, respectively) have three etic indicators, and Guatemala (Figure A2) has three etic indicators and one emic indicator from a household perspective. In Honduras and the Dominican Republic, models utilize data around 2005 justified by a high ecologic correlation of contemporaneous structural indicators on multidimensional poverty from Oxford.

As shown by fitted models on the top of Table 2, MIMIC results converge for the five countries. Generally, assets (AST) and access to essential services (ABS) significantly affect RCI. Adaptive capacity (AC) is considered positive, while social safety nets (SSN) are not recognized as a strategy for improving resilience across those countries. Precisely, ABS and AST are the most relevant pillars across countries, while SSN is positively correlated only in the Dominican Republic (as presented in Table 2).

**Table 2.** MIMIC results by country.

|  | Costa Rica | Dominican Republic | El Salvador | Guatemala | Honduras |
|---|---|---|---|---|---|
| Assets (AST) | 0.2595 *** | 0.5572 *** | 0.2492 *** | 0.1809 *** | 0.3200 *** |
|  | (0.0087) | (0.0260) | (0.0041) | (0.0058) | (0.0154) |
| Access to basic services (ABS) | 0.1339 *** | 0.1095 *** | 0.1207 *** | 0.1686 *** | 0.0261 *** |
|  | (0.0067) | (0.0259) | (0.0035) | (0.0044) | (0.0046) |
| Social safety nets (SSN) | −0.0002 | 0.0613 *** | −0.0135 *** | −0.0326 *** | −0.0075 |
|  | (0.0221) | (0.0042) | (0.0017) | (0.0075) | (0.0048) |
| Adaptive capacity (AC) | 0.0802 *** | 0.0464 *** | 0.1430 *** | −0.0203 *** | 0.1557 *** |
|  | (0.0060) | (0.0082) | (0.0034) | (0.0035) | (0.0190) |
| ABS *AST |  | −0.01791 | 0.0050 * | 0.0025 | −0.0247 *** |
|  |  | (0.0198) | (0.0026) | (0.0056) | (0.0051) |
| AST *AC |  |  |  |  | −0.0986 *** |
|  |  |  |  |  | (0.0099) |
| AC *SSN |  |  |  |  | −0.0001 |
|  |  |  |  |  | (0.0033) |
| Food expenditure per person per day (FEXPPD) | 1 |  |  | 1 | 1 |
|  | (0) |  |  | (0) | (0) |
| Share of food expenditure (Engel) | 0.4340 *** | 1 |  |  |  |
|  | (0.0243) | (0) |  |  |  |
| Negative of the share of SSEXR | 0.2951 *** | 0.3528 *** |  | 0.6768 *** | 0.1317 *** |
|  | (0.0588) | (0.0269) |  | (0.0171) | (0.0040) |

**Table 2.** *Cont.*

|  | Costa Rica | Dominican Republic | El Salvador | Guatemala | Honduras |
|---|---|---|---|---|---|
| Household dietary diversity (HDDS9) |  |  |  | 1.1678 *** (0.0521) | 0.6985 *** (0.0275) |
| The ratio of non-food to food expenditure |  |  | 1 (0) |  |  |
| Total expenditure per person per day |  |  | 1.0807 *** (0.0112) |  |  |
| Food insecurity experience (FIE) (Rasch value of ELCSA) |  |  |  | 5.7913 *** (0.1686) |  |
| Chi−squared ($\chi^2$) | 134.77 | 26.87 | 232.83 | 1050.42 | 131.77 |
| *p*-value | 0.000 | 0.000 | 0.000 | 0.000 | 0.000 |
| TLI | 0.914 | 0.959 | 0.976 | 0.812 | 0.952 |
| CFI | 0.954 | 0.985 | 0.991 | 0.877 | 0.972 |
| RMS | 0.053 | 0.026 | 0.049 | 0.073 | 0.032 |
| Pr RMSEA | 0.246 | 1.000 | 0.589 | 0.000 | 1.00 |
| Number of observations | 5 623 | 8 222 | 23 669 | 11 317 | 8 121 |

Notes: Standard errors in parentheses; *** $p < 0.01$, * $p < 0.10$, two-tailed tests. Source: Authors' own elaboration.

Table 2 shows results separately by country to underline differences in critical determinants of resilience of food and nutrition security (RFNS) among the five countries. Specifically, even if the coefficients of the pillars and food security indicators, and the corresponding interaction (see Figure 1), estimated for all the countries, are generally significant, the relative importance of the pillars (ABS, AST, SSN, and AC) and relevance of indicators of critical determinants (see Figures 2 and 3), as well as the interaction effects, are country-specific.

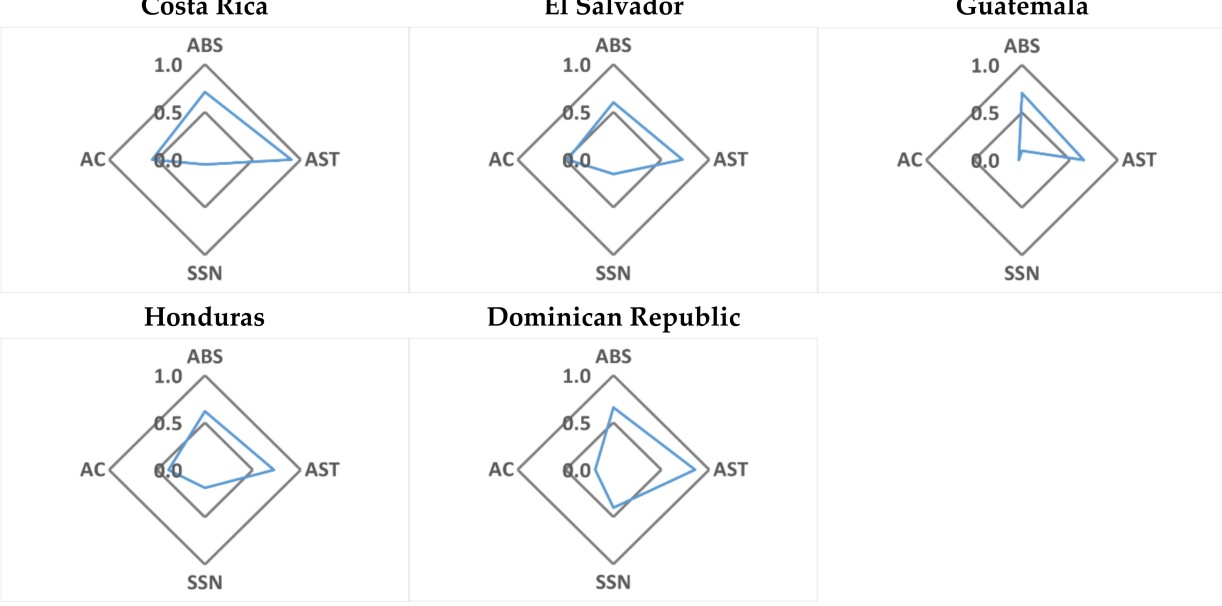

**Figure 1.** Correlation between the resilience of food and nutrition security and key determinants by country. Source: Authors' own elaboration.

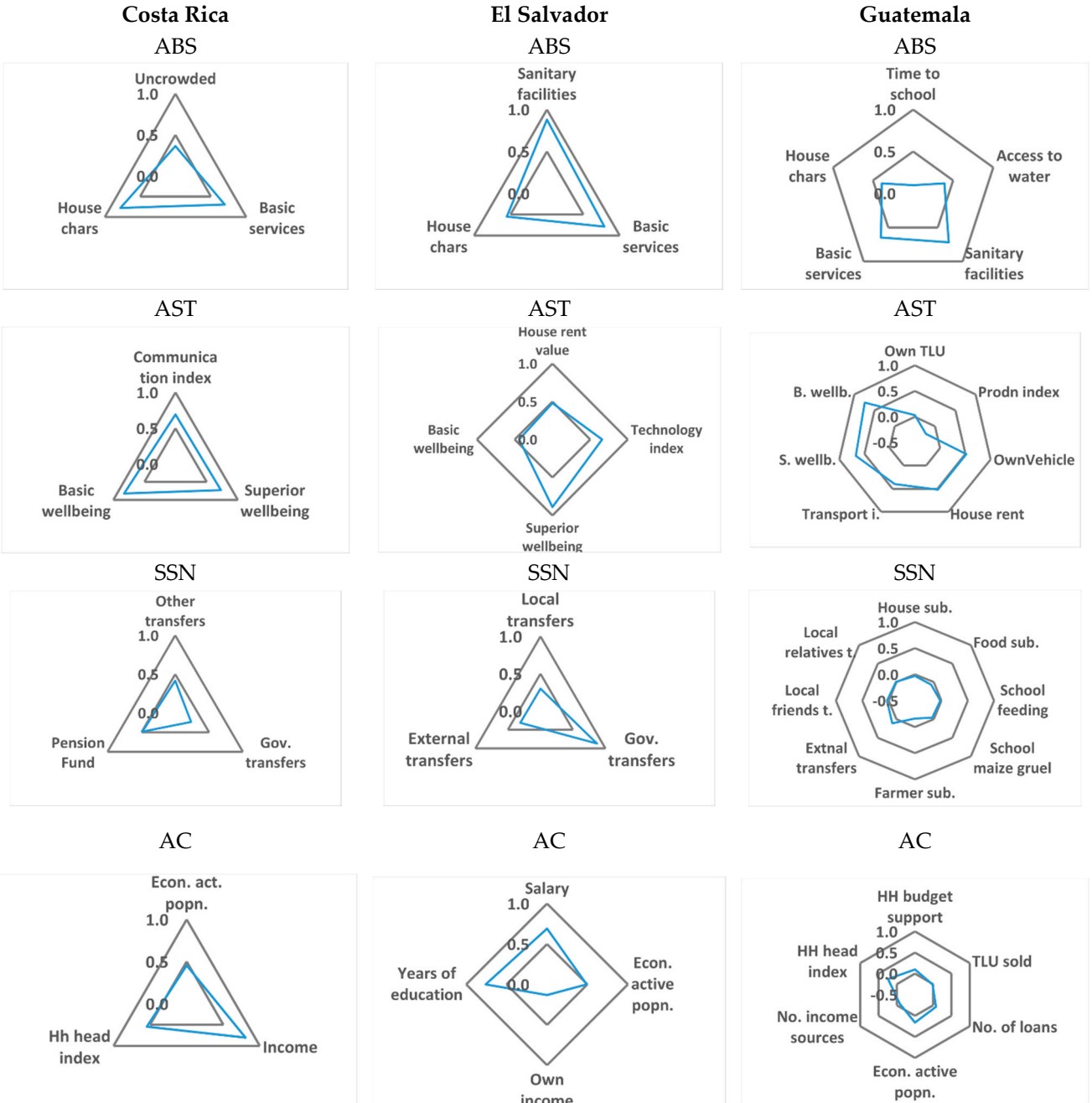

**Figure 2.** Relevance of indicators of critical determinants for Costa Rica, El Salvador, and Guatemala. Source: Authors' own elaboration.

Regarding food and nutrition security, seven indicators are reported in the middle of Table 2. In Guatemala and Honduras, daily food expenditure per person (FEXPPD) is an indicator of food quantity while food quality is measured by the share of staple starchy food expenditure (SSEXR) and additionally food diversity (HDDS9). Costa Rica maintained the share of food expenditure (Engel) as an expression of food quantity. For El Salvador, the anchor indicator is expressed by total spending per person and by the ratio of non-food to food expenditure. In the Dominican Republic, the anchored food quantity is the share of food expenditure, while food quality is indicated by the (negative) percentage of staple starchy food expenditure (SSEXR). Guatemala included adult food insecurity experience (FIES) as part of the food and nutrition security indicators. The FIES refers to and reports about the experiences of the individual respondent or of the respondent's household as a whole on food insecurity. The questions focus on self-reported food-related behaviors

and experiences associated with increasing difficulties in accessing food due to resource constraints. Further details can be found here: http://www.fao.org/policy-support/tools-and-publications/resources-details/en/c/1236494/, accessed on 8 August 2021, calculated like a Rasch transformation of ELCSA).

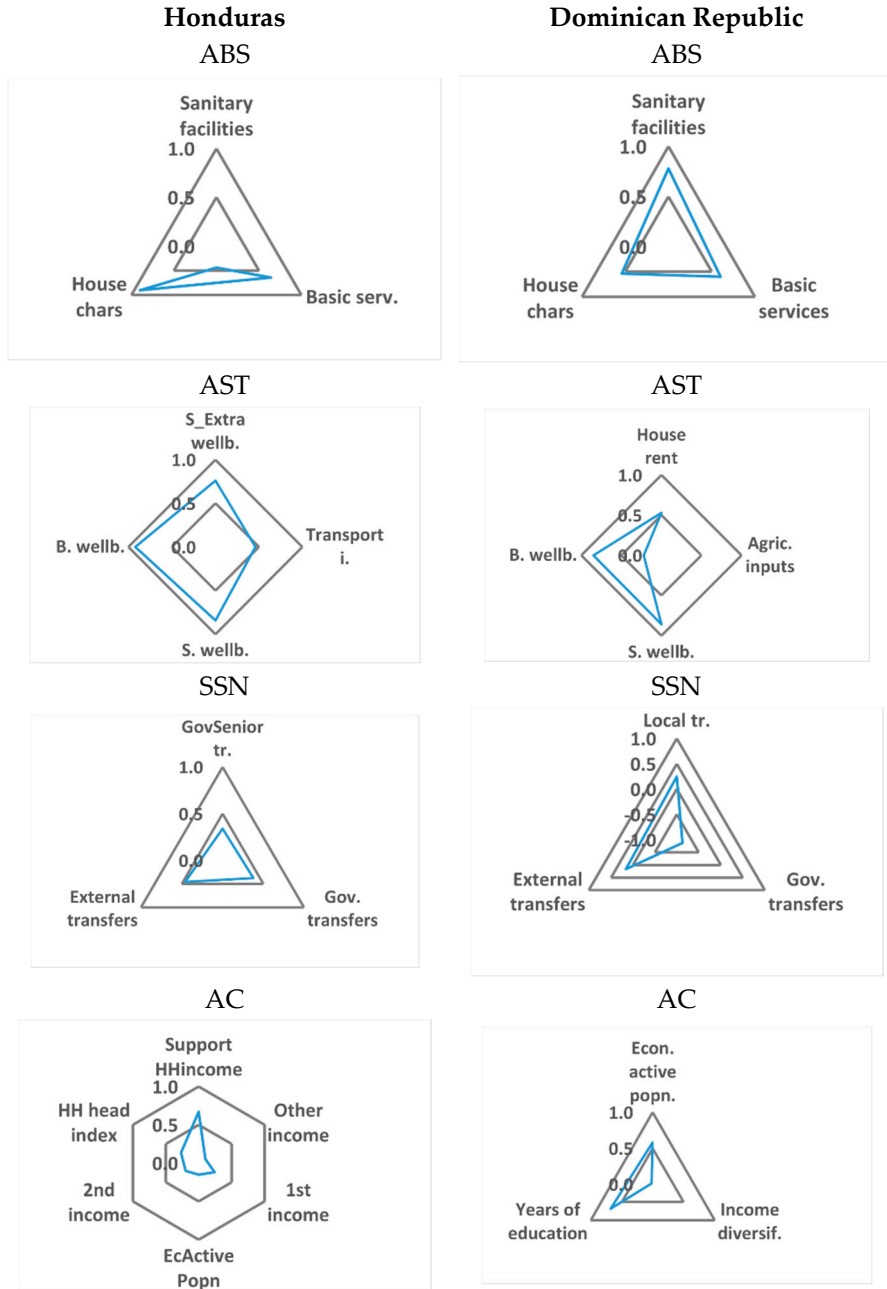

**Figure 3.** Relevance of indicators of critical determinants for Honduras and Dominican Republic. Source: Authors' own elaboration.

As confirmed by the lower section of Table 2, the RIMA-II approach guarantees the acceptable performance of the MIMIC estimation in terms of fit-statistics (Chi2, *p*-value, RMSEA, Pr RMSEA, CFI, and TLI). Indeed, TLI, CFI, and Pr RMSEA are sufficiently consistent for all the countries, even if Guatemala shows a higher level of Chi2 than other countries. The two-step approach allows including many variables relevant for estimating the multidimensional RCI in the different country contexts.

Moreover, statistics on model performances are reported at the bottom of Table 2. El Salvador, Honduras, and the Dominican Republic show good fits with a comparative fit

index (CFI) and Tucker–Lewis index (TLI) greater than 0.90 and a root means square error of approximation (RMSEA) less than 0.05. Costa Rica shows CFI and TLI greater than 0.90, and Guatemala shows RMSEA of <0.073. A large sample size of surveys and disregarded dimensions yield high values of chi-squared and low probabilities.

Even if cross-country analysis shows similarities among the five countries, local specificity remains relevant in the heterogeneous contexts analyzed. Generally, good key determinants are more country-specific than cross-countries.

### 3.2. Cross-Countries Comparison

The principal evidence from the cross-country analysis is that gender is highly relevant. Specifically, single-parent female-headed households are less resilient across countries than traditional male-headed households are, as shown by ordinary least squares (OLS) multiple linear regression in Table 3. In most SICA member countries, particularly in Guatemala, this trend can be explained by comparing female-headed households and male-headed households. Female-headed households have lower average years of education and less ownership or access to assets such as land and credits, livestock, and most inputs and farm tools, as well as access to services that are relevant for productive activities in rural areas [28]. Moreover, across regions, female-headed households also typically operate smaller landholdings than male-headed households, and there are significant and systematic gender differences regarding livestock, financial services, modern inputs, information and extension, and labor [28].

There is significant heterogeneity across regions, countries, locations, and the context in the role of rural women and their participation in agricultural and other economic activities [29]. Despite this heterogeneity, women across regions and contexts face a surprisingly similar set of constraints, which limits their access to productive assets and inputs as well as employment opportunities. While the exact degree of gender inequality in access differs by assets and location, the underlying causes are repeated across contexts: social norms, household/reproductive duties that create time constraints, and asset complementarities (for example having access to land helps with access to credit, which helps with access to purchased inputs). These gender inequalities negatively affect women's productivity and thus involve costs in terms of lost output, income, and ultimately the welfare of households, communities, and nations [28].

Across countries, few indicators show similar relevance for the pillars; in El Salvador, Guatemala, and the Dominican Republic, sanitary facilities are essential services, while house characteristics are critical in Costa Rica, El Salvador, Honduras, and the Dominican Republic. Regarding the similarities for assets, the primary wealth index is generally considerable (relevant in El Salvador), while the index of superior wellbeing is highly significant for all countries. Adaptive capacity (AC) is generally applicable across countries (except in Guatemala), while the social safety nets (SSN) pillar shows significantly positive effects in the Dominican Republic, non-significant effects in Costa Rica and Honduras, and significantly negative effects in El Salvador and Guatemala.

For this reason, the SSN pillar is not necessarily recognized as a strategy for developing resilience across the analyzed Caribbean countries. Notwithstanding, with climate change likely to result in an increased magnitude and frequency of shocks, innovative approaches to social safety nets might be needed to bolster local resilience, support livelihood diversification strategies, and reinforce people's coping strategies in rural areas [30]. Safety net interventions can contribute to agricultural and economic growth by impacting asset creation, asset protection, resource allocation, and redistribution to a woman [31]. If SSNs are invested in productive assets (land and livestock) and directed to women, this can play a significant role in reducing poverty and enhancing resilience while simultaneously empowering women and closing the gender-asset gap [31].

### 3.3. Country Specificity for Heterogeneous Contexts

Critical determinants of resilience in the Central America and the Caribbean countries hold across all conditions but involve a set of variables that depend on local conditions (see the Appendix A, Tables A1–A5).

Table 3 shows that Caribbean countries are heterogeneous in resilience and food security determinants because of the different effects of the household size on resilience (except for Costa Rica and Honduras), the impact of the prevalence of children under 18 years old, and the gender of the household, such as the number of male and female adults as well as female-headed households versus male-headed households. Moreover, Table 3 depicts differences in resilience between rural versus urban households, the household livelihoods, and agricultural versus non-agricultural households. The control groups are characterized by male household heads and large non-poor households living with non-agricultural livelihoods in urban areas for all countries. The reference groups are in the territory of Central Region, department of Ahuachapán, Central District in Francisco Morazán, and National District, for Costa Rica, El Salvador, Guatemala, Honduras, and the Dominican Republic, correspondingly. We also investigated, for Guatemala and Honduras, the resilience difference for households by the mother tongue of household head, and between poor and non-poor households in El Salvador, Guatemala, and the Dominican Republic. Coefficents are not presented in Table 3, since the lack of data from the other 3 countries would have made that table hard to read. They are maternal language: Maya(GT) −4.374 *** (0.254) and non-Spanish(HD) −4.0198 *** (0.6467). Households with the mother tongue of the household head are generally less resilient: for example, Maya versus non-Maya (Spanish, Garifuna, or Xinca) in Guatemala and non-Spanish languages (Garifuna, Miskito, Sumo, Pech, Jicaque, Lenca, Mayangna, Tol, and Mayan) versus Spanish in Honduras. No data are available for other languages different from Spanish for Costa Rica, El Salvador, and the Dominican Republic.

**Table 3.** OLS multiple linear regression for an explanation of the resilience of food and nutrition security in Costa Rica, the Dominican Republic, El Salvador, Guatemala, and Honduras.

| | Costa Rica | Dominican Republic | El Salvador | Guatemala | Honduras |
|---|---|---|---|---|---|
| Intercept | 61.205 *** | 73.864 *** | 50.078 *** | 57.620 *** | 61.1557 *** |
| | (0.673) | (0.356) | (0.380) | (0.425) | (0.6667) |
| Economic shock | | | 0.155 | | |
| | | | (0.152) | | |
| Natural shock (drought/flood) | | | −0.186 | | |
| | | | (0.210) | | |
| Other shocks | | | 1.236 *** | | |
| | | | (0.381) | | |
| Number of children < 5 years | −3.829 *** | −0.337 *** | 0.742 *** | 0.154 | −2.0738 *** |
| | (0.389) | (0.126) | (0.116) | (0.125) | (0.1818) |
| Number of children from 5 to <10 years | −1.657 *** | −1.457 *** | 2.419 *** | −0.338 *** | −1.7965 *** |
| | (0.384) | (0.119) | (0.117) | (0.126) | (0.1706) |
| Number of children from 10 to <18 years | −0.284 | −0.891 *** | 2.935 *** | −0.859 *** | −1.0972 *** |
| | (0.263) | (0.093) | (0.081) | (0.086) | (0.1340) |
| Gender: number of female adults | 5.000 *** | 1.172 *** | 3.794 *** | 0.851 *** | 1.8657 *** |
| | (0.322) | (0.121) | (0.087) | (0.115) | (0.1860) |
| Gender: female household head | −3.714 *** | −0.320 * | −1.738 *** | −1.891 *** | −1.3625 *** |
| | (0.498) | (0.186) | (0.143) | (0.235) | (0.3469) |
| Gender: number of male adults | 1.568 *** | −0.300 *** | 2.177 *** | 0.282 ** | −0.9457 *** |
| | (0.309) | (0.110) | (0.092) | (0.119) | (0.1898) |

**Table 3.** *Cont.*

|  | Costa Rica | Dominican Republic | El Salvador | Guatemala | Honduras |
|---|---|---|---|---|---|
| Livelihood: agricultural |  | −3.131 *** | −2.028 *** | −3. 542 *** | −2.8231 *** |
|  |  | (0.285) | (0.159) | (0.230) | (0.4310) |
| Household size: small |  | −0.985 *** | −3.522 *** | −0.347 | −0.8448 ** |
|  |  | (0.243) | (0.189) | (0.252) | (0.3971) |
| Poverty: extreme (food poor) |  | −13.692 *** | −16.254 *** | −18.500 *** |  |
|  |  | (0.210) | (0.218) | (0.304) |  |
| Poverty: poor |  | −6.767 *** | −9.864 *** | −11.738 *** |  |
|  |  | (0.199) | (0.140) | (0.210) |  |
| Area: rural | −7.719 *** | −3.574 *** | −7.971 *** | −5.326 *** | −12.3324 *** |
|  | (0.459) | (0.186) | (0.140) | (0.195) | (0.3785) |

Notes: Standard errors in parentheses; *** $p < 0.01$, ** $p < 0.05$, * $p < 0.10$, two-tailed tests. Source: Authors' own elaboration.

Heterogeneity is relevant both among and within the five countries, as Table 4 shows. In Costa Rica, households from the Central region are more resilient than the other five regions [32]. In El Salvador, households from the territory of Ahuachapán are less resilient than households from the other 13. In Guatemala, the benchmark region is Sololá and the other 20 regions are less resilient. In Honduras, households in the Central District are less resilient than families in the territory of Islas de la Bahía (similarly to households from the territory of Gracias a Dios, Atlántida, and the territory of Cortés, excluding its capital city of San Pedro Sula), and they are more resilient than in from the remaining 15 territories, including San Pedro Sula [33]. In the Dominican Republic, households from the National District are similar to those from La Altagracia province but more resilient than other provinces from 30 provinces [34].

**Table 4.** OLS multiple linear regression for an explanation of the resilience of food and nutrition security by territory.

| Costa Rica | |
|---|---|
| **Territory (Region)** | |
| Chorotega | −8.817 *** |
|  | (0.678) |
| Pacífico Central | −8.010 *** |
|  | (0.711) |
| Brunca | −9.858 *** |
|  | (0.680) |
| Huetar Caribe | −11.561 *** |
|  | (0.688) |
| Huetar Norte | −9.674 *** |
|  | (0.680) |
| Observations | 5 623 |
| R-squared | 0.2186 |
| **Dominican Republic** | |
| **Territory (Province)** | |
| Azua | −5.798 *** |
|  | (0.522) |
| Baoruco | −8.458 *** |
|  | (0.691) |
| Barahona | −7.182 *** |
|  | (0.497) |
| Dajabon | −6.332 *** |
|  | (1.220) |

**Table 4.** *Cont.*

| Dominican Republic | |
|---|---|
| **Territory (Province)** | |
| Duarte | −2.302 *** |
| | (0.577) |
| Elias Piña | −10.115 *** |
| | (0.824) |
| El Seibo | −4.972 *** |
| | (0.582) |
| Espaillat | −1.212 * |
| | (0.624) |
| Independencia | −7.942 *** |
| | (0.882) |
| La Altagracia | 0.102 |
| | (0.474) |
| La Romana | −0.890 * |
| | (0.465) |
| La Vega | −2.286 *** |
| | (0.509) |
| María Trinidad Sánchez | −1.259 * |
| | (0.708) |
| Monte Cristi | −3.659 *** |
| | (0.778) |
| Pedernales | −10.175 *** |
| | (1.297) |
| Peravia | −2.669 *** |
| | (0.553) |
| Puerto Plata | −1.820 *** |
| | (0.541) |
| Hermanas Mirabal | −3.410 *** |
| | (0.922) |
| Samana | −2.835 *** |
| | (0.854) |
| San Cristóbal | −2.092 *** |
| | (0.395) |
| San Juan | −6.683 *** |
| | (0.472) |
| San Pedro de Macorís | −2.333 *** |
| | (0.423) |
| Sánchez Ramírez | −3.450 *** |
| | (0.724) |
| Santiago | −1.595 *** |
| | (0.408) |
| Santiago Rodríguez | −3.713 *** |
| | (1.085) |
| Valverde | −1.722 ** |
| | (0.731) |
| Monseñor Nouel | −2.576 *** |
| | (0.701) |
| Monte Plata | −6.920 *** |
| | (0.467) |
| Hato Mayor | −6.628 *** |
| | (0.595) |
| San José de Ocoa | −4.085 *** |
| | (0.832) |
| Santo Domingo | −1.910 *** |
| | (0.350) |
| Observations | 8 222 |
| R-squared | 0.5822 |

**Table 4.** *Cont.*

| El Salvador | |
|---|---|
| **Territory (Departamento)** | |
| Santa Ana | 2.261 *** |
| | (0.311) |
| Sonsonate | 0.901 *** |
| | (0.343) |
| Chalatenango | 4.483 *** |
| | (0.407) |
| La Libertad | 3.380 *** |
| | (0.321) |
| San Salvador | 3.985 *** |
| | (0.299) |
| Cuscatlán | 1.256 *** |
| | (0.328) |
| La Paz | 1.143 *** |
| | (0.358) |
| Cabañas | 1.914 *** |
| | (0.352) |
| San Vicente | 2.096 *** |
| | (0.360) |
| Usulután | 1.548 *** |
| | (0.388) |
| San Miguel | 3.501 *** |
| | (0.326) |
| Morazán | 1.486 *** |
| | (0.355) |
| La Unión | 3.539 *** |
| | (0.352) |
| Observations | 23 670 |
| R-squared | 0.4967 |
| **Guatemala** | |
| **Territory (Departamento)** | |
| El Progreso | −5.568 *** |
| | (0.508) |
| Sacatepéquez | −1.413 *** |
| | (0.422) |
| Chimaltenango | −2.861 *** |
| | (0.507) |
| Escuintla | −5.230 *** |
| | (0.446) |
| Santa Rosa | −5.490 *** |
| | (0.541) |
| Sololá | −0.277 |
| | (0.561) |
| Totonicapán | −2.400 *** |
| | (0.560) |
| Quetzaltenango | −2.617 *** |
| | (0.453) |
| Suchitepéquez | −6.091 *** |
| | (0.468) |
| Retalhuleu | −5.582 *** |
| | (0.541) |
| San Marcos | −2.800 *** |
| | (0.542) |
| Huehuetenango | −2.268 ** |
| | (0.536) |
| Quiché | −2.494 *** |
| | (0.556) |

**Table 4.** *Cont.*

| Guatemala | |
|---|---|
| **Territory (Departamento)** | |
| Baja Verapaz | −5.793 *** |
| | (0.574) |
| Alta Verapaz | −5.948 *** |
| | (0.592) |
| Petén | −6.495 *** |
| | (0.486) |
| Izabal | −1.397 *** |
| | (0.554) |
| Zacapa | −1.121 *** |
| | (0.556) |
| Chiquimula | −5.527 *** |
| | (0.566) |
| Jalapa | −5.498 *** |
| | (0.519) |
| Jutiapa | −6.459 *** |
| | (0.475) |
| Observations | 11 317 |
| R-squared | 0.583 |
| **Honduras** | |
| **Territory (Departamento)** | |
| Atlántida | −0.3244 |
| | (0.8210) |
| Colón | −2.9900 *** |
| | (0.7058) |
| Comayagua | −6.5267 *** |
| | (0.8377) |
| Copán | −11.0965 *** |
| | (0.7381) |
| San Pedro Sula, Cortés | −1.2310 *** |
| | (0.3037) |
| Resto de Cortés | 0.8205 |
| | (0.6343) |
| Choluteca | −6.7883 *** |
| | (0.7879) |
| El Paraíso | −12.5532 *** |
| | (0.6816) |
| Resto de Francisco Morazán | −6.4972 *** |
| | (1.0356) |
| Gracias a Dios | 0.5682 |
| | (1.9858) |
| Intibucá | −7.7107 *** |
| | (0.9957) |
| Islas de la Bahía | 6.5423 *** |
| | (1.9621) |
| La Paz | −7.6270 *** |
| | (0.9261) |
| Lempira | −12.1412 *** |
| | (1.0688) |
| Ocotepeque | −8.2790 *** |
| | (1.1276) |
| Olancho | −6.3465 *** |
| | (0.6445) |
| Santa Bárbara | −10.7710 *** |
| | (0.6556) |

**Table 4.** *Cont.*

| Honduras | |
|---|---|
| **Territory (Departamento)** | |
| Valle | −8.7641 *** |
| | (1.1645) |
| Yoro | −2.5708 *** |
| | (0.7092) |
| Observations | 8 123 |
| R-squared | 0.4301 |

Notes: Standard errors in parentheses; *** $p < 0.01$, ** $p < 0.05$, * $p < 0.10$, two-tailed tests. Source: Authors' own elaboration.

Additionally, households with more young (<5 years), primary-school-aged (5 to <10 years), and secondary-school-aged children (10 to <18 years) are generally more resilient in El Salvador [35]. In contrast, Honduras and the Dominican Republic show less resilience for households with young, primary-school-aged, or secondary-school-aged children. In Costa Rica, households with more young and primary-school-aged children are less resilient than other groups, while no difference occurs between households with or without secondary-school-aged children. In Guatemala, households with more primary- or secondary-school-aged children are less resilient than other groups, while no difference occurs between households with or without more young children [36]. In general, small households are less resilient to food and nutrition insecurity in El Salvador, Guatemala, and the Dominican Republic, even if large households show one point lower than small-sized households in the Dominican Republic. No difference in resilience occurs for household size in Guatemala and the Dominican Republic. No data are available for Honduras and Costa Rica.

Country specificity, such as the heterogeneity of territories and household composition, affects the nature and magnitude of resilience. Figures 2 and 3 show that key determinants are not equally significant across countries. For instance, the wealth index and superior or extra-superior wellbeing of assets are relevant across countries but based explicitly on different investments. Extra excellent wellbeing of assets is applicable only in Honduras, where transportation is significant; communication is essential for Costa Rica and house rental for the Dominican Republic. Moreover, highly relevant dimensions of the AC pillar are years of education in El Salvador and the Dominican Republic, income or salary in Costa Rica and El Salvador, support to income in Honduras, and economically active population in the Dominican Republic (to less extent in Costa Rica and El Salvador). Pensions and other transfers are relevant in Costa Rica, international remittances are relevant in Honduras, while governmental transfers are relevant in El Salvador and the Dominican Republic. Moreover, the negative value estimated in the Dominican Republic reflects targeting households with very low resilience to food and nutrition insecurity.

According to contexts and households' needs, the mentioned results provide data-driven evidence on the market for context-specific resilience measurement and analysis to properly diversify and improve food security and resilience-enhancing interventions. Policy designers, and, and decision-makers should consider the peculiarities of each country and household composition to enhance the corresponding critical determinants of RFNS nationwide or at territorial levels. For these reasons, even if a detailed description of the implications is omitted in this paper, technical reports should be comprehensive, containing details by territories and household type, so that similarities can be identified for national public policies and specific relationships for territorial interventions for improving key determinants and enhancing RFNS.

### 3.4. Farm Diversification

Rural households relying on agricultural livelihoods are less resilient to food and nutrition insecurity than non-agricultural ones across the five countries, except Costa Rica,

due to no data being available. As Table 3 shows, there is evidence from OLS multiple regression estimation of a negative resilience level for rural and vulnerable areas, especially those exposed to drought for Costa Rica, Dominican Republic, and El Salvador. Usual household diets in rural areas of low-income countries are often limited to one or two starchy staple foods and may be especially lacking in micronutrient-rich fruits, vegetables, and animal-source foods.

Farm diversity is measured by the number of different crops cultivated during the dry season, distribution of the area of land owned, and the number of livestock species reared by households [37]. For farming households that primarily consume what they produce, it seems reasonable that diversified agricultural production would lead to more diverse diets. More diverse production systems contribute to more varied household diets that, in turn, positively influence the nutritional status and resilience of household members [37].

The MIMIC result shows the strong relationship between resilience and household dietary diversity (HDDS). Household dietary diversity is a highly context-specific indicator [38] and is positively influenced by farm production diversity. Specifically, the MIMIC result shows the high relevance of food quality and resilience; for example, Guatemala, followed by the Dominican Republic and Costa Rica, experienced higher relationships of resilience and food quality (negative of the share of staple starchy food expenditure, SSEXR), while marginally in Honduras.

Nationwide findings on food and nutrition security indicators, depicting food consumption quantity and quality and the positive relation of resilience and dietary diversity in rural areas, as Table 2 shows, provide inputs to direct interventions. Policy actions should increase the variety of food with local products through comprehensive farming practices, allowing micronutrient sources accessible for different groups of households in the Caribbean countries. Specifically, actions prioritizing the institutional supply of inputs for crop production and yield, and agricultural and livestock diversification, such as small livestock farming for sale on the local market and crops with minimal water needs, are relevant for resilience-enhancing strategies in rural areas. Farm diversification improves household dietary diversity, which results in higher resilience to food insecurity in the five countries.

Finally, improved access to education and training programs on farming practices and techniques offer the basic knowledge that positively influences crop and livestock production diversity resistance to drought; farm diversity promotes job and income diversification sources across countries.

## 4. Conclusions and Policy Implications

Policies call for entry points that allow programmatic actions identifiable in key determinants as causes and food and nutrition security indicators as responses. As some key determinants result from others and depend on the context, policies should address dimensions of key determinants coherent to resources and local settings.

Most likely, the main entry points call for strengthening dimensions from the household adaptive capacity (AC), with or without measurements from private or public social safety nets (SSN), and dimensions aiming at the improvement of access to essential services (ABS) and increasing household assets (AST), targeting household groups with low resilience.

Enhancing adaptive capacity (AC) involves taking advantage of dimensions depicting positive relation with RFNS, targeting territories, and house groups with low levels of those dimensions or low RFNS. Several actions should improve the knowledge and practice of successful agricultural experiences (crops, livestock, aquaculture, and fishery) to better diversify food production and food availability of farm households and non-agricultural households in local markets. It involves the training of economically active populations for utilizing technologies and collaborative strategies for increasing agricultural productivity and farming diversity, making available a higher food consumption diversity (HDDS). It includes training household member caregivers to incorporate more comprehensive

options of food adequate in quantity and quality for food consumption patterns, taking into consideration local culture in preferences and preparation of food, especially for young children, lactating women, and senior household members. As income is one of the standard dimensions with high correlation, income-generating diversified activities are needed in all countries, strengthening education for adults in El Salvador, Guatemala, and the Dominican Republic and formal education for youngsters in Guatemala and the Dominican Republic.

Policies supporting social safety nets (SSN) have two main objectives. The core objective is not resilience but short-term relief to crises or emergencies, in particular, vulnerable families with inputs for the consumption of basic needs such as food and health; for example, pension funds should keep up with inflation rates in Costa Rica [39], and governmental transfers should target households in need in El Salvador and the Dominican Republic ([40,41]). The second objective is to improve the adaptive capacity (AC) with productive inputs for providing eroded assets and livelihoods, for example, helping the effective use of external remittances in all countries except Costa Rica.

Policies supporting access to essential services (ABS) are sort of transfers in kind to the population and subsidized services to vulnerable people, for example, electricity to rural households. Investments in making more accessible essential services to the population increase food security in all countries; however, higher returns are mainly in essential services (electricity, solid disposal, and sewage) in all countries, excepting the Dominican Republic; in improving sanitation services in El Salvador, Guatemala, and the Dominican Republic; and in supporting housing improvements in Costa Rica and Honduras.

Policies for improving assets (AST) for a better RFNS are costly; however, by supporting the other key determinants simultaneously, assets may build up, and resources may become more available so that resilience in the sense of being prepared for short-term shocks is improved. Policies tackling the building up of assets (AST) call for education and orientation of diversified programs targeting rural populations, in particular, smallholder farmers for the promotion of autochthonous seed production, production, and utilization of organic fertilizers and natural pesticides for staple food crop production and complement with yellow and green vegetables as well as yellow fruits (rich in iron, beta-carotene, and vitamin C), yellow roots, and tubers (rich in beta-carotenes and carbohydrates). It also includes local small livestock reproductive units for an enhanced production for own-production consumption such as eggs and poultry. It also includes selling shows in local markets for income-generating purposes and complementing food consumption diversity by purchasing food such as salt fortified with iodine and fluoride, sugar fortified with vitamin A, vegetable oil, and other non-locally produced food items.

Policies tackling dimensions measured by food and nutrition security indicators should be sensitive to resilient food systems. Resilient food systems are based on agriculture and small livestock activities oriented first toward increasing resilience to food and nutrition insecurity by improving the quantity (FEXPPD or EngelR) and quality of dietary patterns with more diversified food consumption (SSEXR and HDDS) from own-production, including local agro-industry and second toward generating income for meeting food needs from local markets (FEXPPD) and non-food needs (health, education, communications, etc.).

For example, diversity in food may increase the demand for cheese and other dairy products, household bakeries using local inputs such as eggs, maize flour, and dry legume flour (soybeans or pigeon pea or chickpea, or any available legume), and vegetables and fruits as well as roots and tubers. Actions should aim at improving the food and nutritional status of young children and women in pregnancy or lactation, for example, promoting the consumption and production of optimized food mixtures in essential amino acids based on local recipes. By adding non-starchy food to the local diet, the share of staple starchy food (SSEXR) may lower and increase dietary quality, reaching macronutrient contributions to total energy as recommended by experts.

The increasing lack of diversity in farming and food systems is one of the greatest threats to long-term sustainability. Climate-related shocks are key push factors for diversification [42,43]. In the context of food insecurity and risk of the resilience of agri-food systems under climate change and land degradation, farm diversity is a flexible approach to avoid such shocks ex-ante [44]. Livelihood diversification strategies, including crop and income diversification, are fundamental in these contexts [43,45]. Diversified farming implies farms that integrating several crops and animals in the production system and promote agrobiodiversity across scales, regenerating ecosystem services and reducing the need for external inputs [46]. Specifically, approaching diversity at the farm level stimulates technology, information, and knowledge [47], which can be used by farmers to cope with current or future challenges, reducing vulnerability [48] and improving adaptive capacity [49].

Generally, smallholder farming systems are highly heterogeneous in many characteristics such as land access, soil fertility, cropping, livestock assets, off-farm activities, labor and cash availability, socio-cultural traits, farm development trajectories, and livelihood orientations [50]. Cross-countries evidence in Central America shows that small farmers, using diversification practices such as cover crops, inter-cropping, and agroforestry, suffered more minor damage than conventional monoculture neighbors after extreme climatic events [51]. In particular, there are an estimated 17 million family farms in Latin American and the Caribbean, which represent around 60 million people, 80 percent of all farms, and 35 percent of the cultivated land in the region. Family farming contributes 40 percent of total agricultural output and generates over 60 percent of jobs related to agriculture in the area [52].

Country evidence shows different sensitivity to food and nutrition security indicators. For example, food quantity (FEXPPD) more than food quality (SSEXR) and inequality in food quantity (for Costa Rica, El Salvador, and the Dominican Republic) should be addressed. Guatemala and Honduras should address both food quantity (FEXPPD) and food quality (SSEXR). Making sure that the right quantity of food reaches low-income population groups becomes crucial, accompanied by comprehensive nutritional education on how food quality can be achieved or maintained. Especially in Guatemala and Honduras, we advise not only addressing food quality and food quantity but also providing comprehensive nutritional education and farm diversification training for improving food consumption patterns [53,54].

It would be a mistake to consider only key determinants of FNSI, dimensions of food, and nutrition security. It should be part of any policy.

As a possible way forward for this analysis, we think that additional rounds of data would better understand the changes and dynamics of resilience and food security in the region. Additionally, we believe that adopting better indicators for social safety nets (such as remittances and transfers, currently present in few countries only) would allow a deeper evaluation of the essential role of social cohesion and social protection.

**Author Contributions:** Conceptualization, M.d., P.P.d.F. and R.S.; methodology, R.S. and M.d.; software, R.S.; validation, M.d. and P.P.d.F.; formal analysis, R.S.; investigation, F.B.-M.; resources, R.S.; data curation, R.S.; writing—original draft preparation, R.S. and P.P.d.F.; writing—review and editing, M.d. and F.B.-M.; visualization, R.S.; supervision, M.d. and P.P.d.F.; project administration, P.P.d.F.; funding acquisition, P.P.d.F. and R.S. All authors have read and agreed to the published version of the manuscript.

**Funding:** This research was funded by the European Commission under agreement DCI/ALA/2019/411-534.

**Institutional Review Board Statement:** Ethical review and approval were waived for this study, due to secondary utilization and processing data already collected and provided by National Statistics Offices.

**Informed Consent Statement:** Not applicable.

**Data Availability Statement:** Not applicable.

**Acknowledgments:** This paper was prepared by Marco d'Errico and Flavia Benedetti-Michelangeli, from the Agricultural Development Economics Division (FAO), and Ricardo Sibrian and Patricia Palma de Fulladolsa, of the General Secretariat of the Central American Integrating System (PROGRESAN-SICA II). We would like to thank Emiliano Magrini, Rebecca Pietrelli, Jeanne Pinay (ESA division, FAO), Aura Estela Leiva Prado, and Sidia Lisetht Lopez Villalta (PROGRESAN-SICA II) for their inputs and contributions. We would also like to thank Daniela Verona for her support in the publication and editing process.

**Conflicts of Interest:** The authors declare no conflict of interest.

## Appendix A

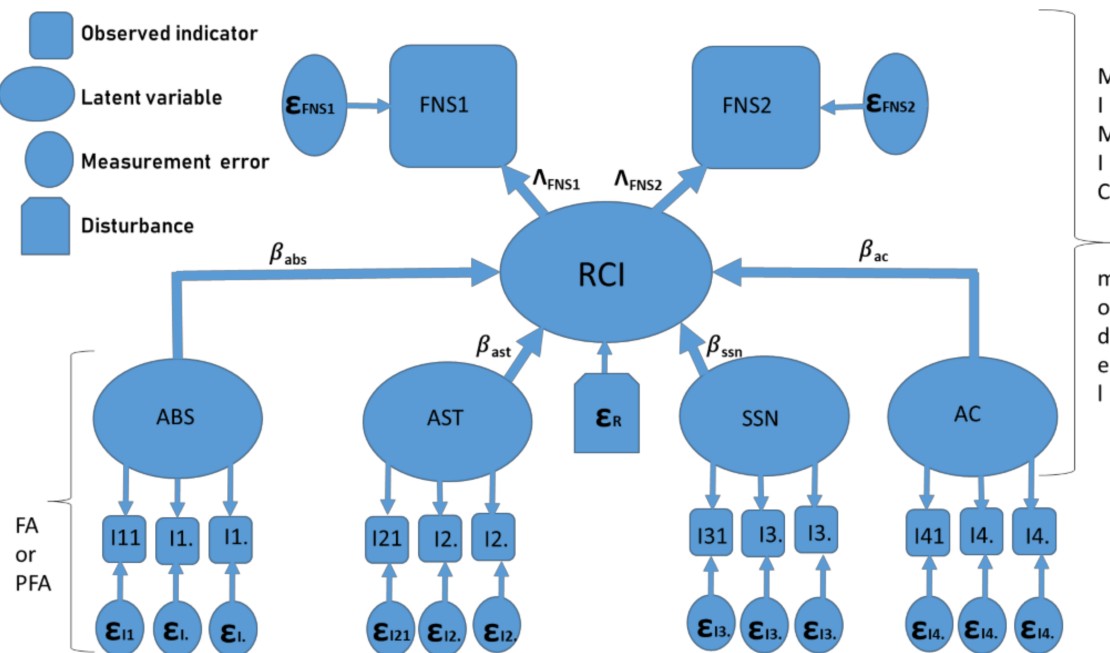

**Figure A1.** RIMA-II MIMIC model with two food and nutrition security indicators. Source: Authors' own elaboration.

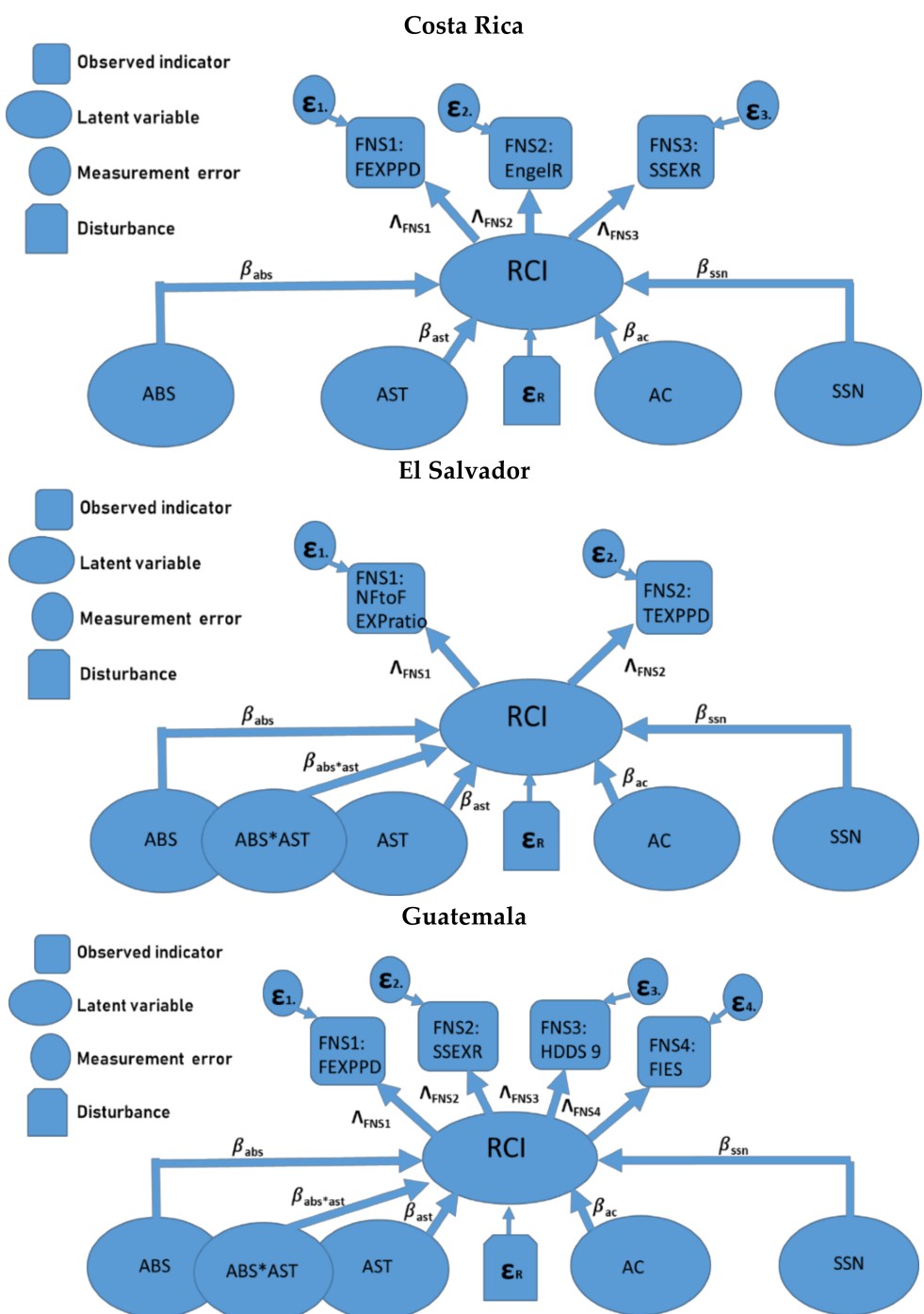

**Figure A2.** RIMA-II MIMIC models developed for Costa Rica, El Salvador, and Guatemala. Source: Authors' own elaboration.

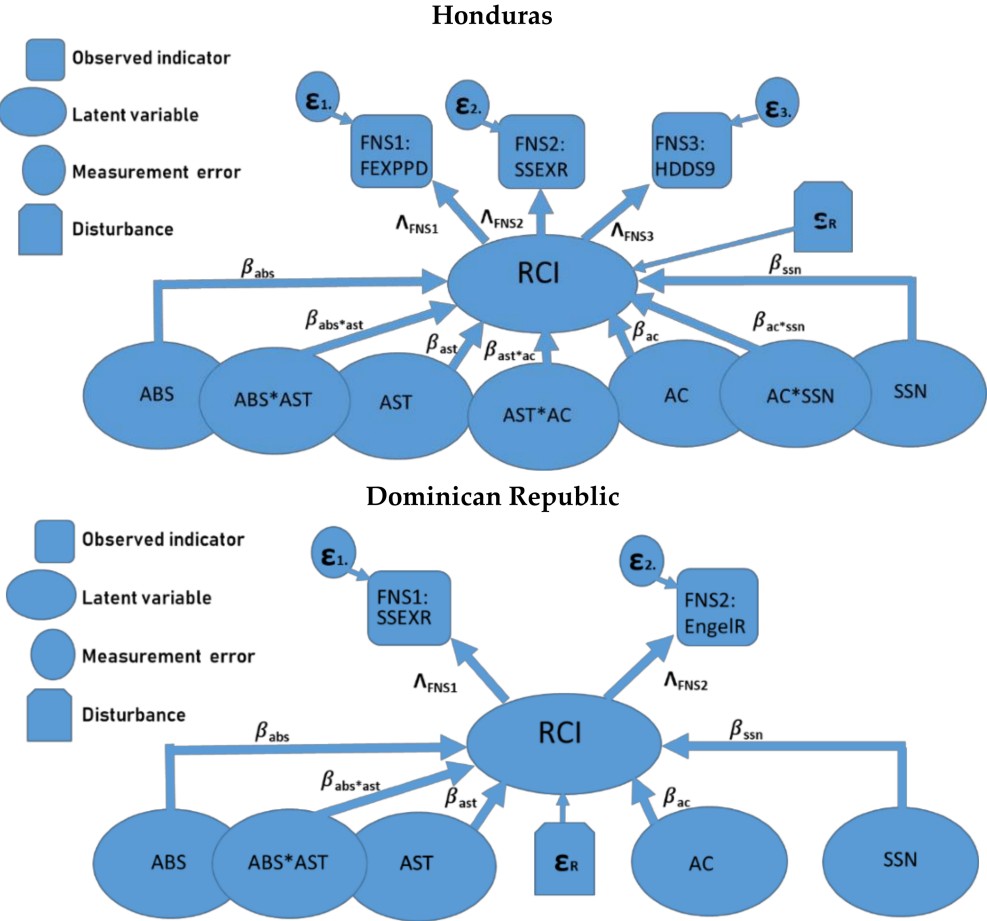

**Figure A3.** RIMA-II MIMIC models developed for Honduras and the Dominican Republic. Source: Authors' own elaboration.

**Table A1.** Indicators for key determinants and dimensions of the resilience of food and nutrition security in Costa Rica.

| Costa Rica | | |
|---|---|---|
| **Key Determinant** | **Variable Code** | **Description** |
| ABS | d_walls | Acceptable wall type |
| | d_floor | Acceptable floor type |
| | d_hhtype | Household type |
| | d_rooms | Number of rooms |
| | hh_characteristics_index | Household characteristics index |
| | d_water | Acceptable water-supply type |
| | d_electricty | Acceptable electricity |
| | d_trash | Acceptable solid disposal |
| | basic_service_index | Basic services index |
| | d_sewer_sys | Acceptable sewer system |
| | d_trwater | Acceptable water treatment |
| | d_latrIne | Acceptable toilet type |
| | d_overcrowding | The acceptable number of people per room |
| | health_house_index | Health house index |

**Table A1.** *Cont.*

| Costa Rica | | |
|---|---|---|
| **Key Determinant** | **Variable Code** | **Description** |
| AST | d_radio | Radio presence |
| | d_tv | Tv presence |
| | d_licuadora | Blender presence |
| | d_maqcoser | Sewing machine |
| | d_plancha | Iron presence |
| | d_microonda | Microwave presence |
| | riquezahogar_index | Wealth index (minor value) |
| | d_house | Dwelling rent |
| | d_comp | Computer presence |
| | d_vehiculo | Automobile presence |
| | d_refri | Refrigerator presence |
| | d_lavadora | Washing-machine presence |
| | riquezahogarextra_index | Extra wealth index (principal value) |
| | d_phone | Telephone presence |
| | d_cell | Mobile phone presence |
| | d_cable | Cable presence |
| | tecnologia_index | Technology Index |
| | cost house | Monthly dwelling rent |
| SSN | d_gas | Gas subsidy |
| | d_elect | Electricity subsidy |
| | d_pagricola | Agriculture inputs presence |
| | gobtransfer_index | Government transfers Index |
| | st_ayudalocal | Remittances from friends or neighbors |
| | st_ayudaexterior | External remittances from family |
| AC | tdep | Income from dependent work |
| | otingdep | Other income from dependent work |
| | tsec | Secondary work |
| AC | ingresos_dep_index | Dependent labor earnings index |
| | tinder | Monthly dependent work income |
| | actnl | Monthly income from non-work activities |
| | actag | Income from agricultural activities |
| | ingresos_indep_index | Independent work income index |
| | log_edavg | Average years of education of members of the household |
| | economic_active | The inverse of the dependency ratio |
| FS | log_gastohog | Log of Expenditure per person per day |
| | ln_rzngnoaligali | The inverse of the proportion of food expenditure to total expenditure |
| | ln_rzngnoaligali | Log of the inverse of the proportion of starch food expenditure to the total food expenditure |

Source: Authors' own elaboration.

**Table A2.** Indicators for key determinants and dimensions of the resilience of food and nutrition security in El Salvador.

| El Salvador | | |
|---|---|---|
| **Key Determinant** | **Variable Code** | **Description** |
| ABS | d_walls | Acceptable wall type |
| | d_floor | Acceptable floor type |
| | d_hhtype | Household type |
| | d_rooms | Number of rooms |
| | hh_characteristics_index | Dwelling characteristics index |
| | d_water | Acceptable water-supply type |
| | d_electricty | Acceptable electricity |
| | d_trash | Acceptable solid disposal |
| | basic_service_index | Basic services index |
| | d_sewer_sys | Acceptable sewer system |
| | d_trwater | Acceptable water treatment |
| | d_latrine | Acceptable toilet type |
| | d_overcrowding | The acceptable number of people per room |
| | health_house_index | Health house index |
| AST | d_radio | Radio presence |
| | d_tv | Television-set presence |
| | d_licuadora | Blender presence |
| | d_maqcoser | Sewing machine |
| | d_plancha | Iron presence |
| | d_microonda | Microwave presence |
| | riquezahogar_index | Household wealth index (minor value) |
| | d_house | Dwelling rent |
| | d_comp | Computer presence |
| | d_vehiculo | Automobile presence |
| | d_refri | Refrigerator presence |
| | d_lavadora | Washing-machine presence |
| | riquezahogarextra_index | Extra wealth index (principal value) |
| | d_phone | Phone presence |
| | d_cell | Mobile phone presence |
| | d_cable | Transmission cable presence |
| | tecnologia_index | Technology Index |
| | cost house | Monthly dwelling rent |
| SSN | d_gas | Gas subsidy |
| | d_elect | Electricity subsidy |
| | d_pagricola | Agriculture inputs |
| | gobtransfer_index | Subsidy index |
| | st_ayudalocal | Remittances from friends or neighbors |
| | st_ayudaexterior | External remittances from family |
| AC | tdep | Income from dependent work |
| | otingdep | Income from other dependent work |
| | tsec | Secondary work |
| AC | ingresos_dep_index | Dependent labor earnings index |
| | tinder | Monthly income from self-employment |
| | actnl | Monthly income from non-work activities |
| | actag | Income from agricultural activities |
| | ingresos_indep_index | Independent work income index |
| | log_edavg | Average household schooling |
| | economic_active | The inverse of the dependency ratio |

**Table A2.** *Cont.*

| El Salvador | | |
|---|---|---|
| **Key Determinant** | **Variable Code** | **Description** |
| FS | log_gastohog | Log of total expenditure |
| | ln_rzngnoaligali | Log of non-food to food expenditure ratio |

Source: Authors' own elaboration.

**Table A3.** Indicators for key determinants and dimensions of the resilience of food and nutrition security in Guatemala.

| Guatemala | | |
|---|---|---|
| **Key Determinant** | **Variable Code** | **Description** |
| ABS | d_hhtype | Acceptable dwelling type |
| | d_floor | Acceptable floor type |
| | d_walls | Acceptable wall type |
| | d_rooms | Acceptable number of rooms |
| | d_kitchen | Acceptable location of the kitchen |
| | hh_characteristics_index | Dwelling characteristics index |
| | d_water | Acceptable water-supply type |
| | d_electricty | Electricity presence |
| | d_trash | Acceptable solid disposal |
| | hh_servbasic_index | Basic services index |
| | d_trwater | Acceptable water treatment |
| | d_latrine | Toilet presence |
| | d_overcrowding | Acceptable number of people per room |
| | d_lena_chim | Acceptable use of wood and chimney |
| | health_service_index | Health service index |
| | Acceso a agua | Closeness to the source of water in meters (inverse distance) |
| | Acceso an escuela | Closeness to school in minutes (inverse distance) |
| AST | d_gasstove | Stove presence |
| | d_fridge | Refrigerator presence |
| | d_washingmach | Washing-machine presence |
| | d_tv | Television-set presence |
| | d_computer | Computer presence |
| | d_celular | Cell-phone presence |
| | d_Internet | Internet-service presence |
| | basic_wealth_index | Primary wealth index (high monetary value) |
| | d_soundsys | Sound-system presence |
| | d_blender | Blender presence |
| | d_pressurec | Pressure-cooker presence |
| | d_iron | Iron presence |
| | extra_wealth_index | Extra wealth index (low monetary value) |
| | d_moto | Motorcycle presence |
| | d_bicycle | Bicycle presence |
| | d_auto | Automobile presence |
| | transport_index | Transport index |
| | tenencia_vehic | Transport ownership |
| | cosecho_prod | Crop harvesting |
| | crio_animales | Livestock raising |
| | Tiene_negocios | Business ownership |
| | Production_index | Production index |
| | ltut | Ownership of livestock tropical units |
| | house_value | Dwelling rent |

**Table A3.** *Cont.*

| Guatemala | | |
|---|---|---|
| **Key Determinant** | **Variable Code** | **Description** |
| SSN | st_ayudafam | Remittances from relatives |
| | st_ayudaper | Remittances from friends or neighbors |
| | st_ayudarem | International remittances |
| | st_iagricolas | Agriculture inputs |
| | st_vasoatol | Cereal gruel food aid program |
| | st_alescolar | School food aid program |
| | st_subsidio_alim | Food subsidy |
| | st_subsidio_viv | Dwelling subsidy |
| AC | edhd | Years of education of household head |
| | ywg | Household head working |
| | healthhd | Household head working affiliated to national health social institution |
| | hh_head_index | Household head capacity index |
| | d_isemd | Weekly income as an employee |
| | d_isemi | Weekly income as an entrepreneur |
| | d_isemiag | Weekly income as an agricultural entrepreneur |
| | participation_index | Participation in household income index |
| | economic_active | The inverse of dependency ratio |
| | nincsrc | Number of income sources |
| | num_prestamos | Number of loans |
| | negtluv | Selling of livestock tropical units (hostile) |
| FS | log_fexppd | Log of daily expenditure per person |
| | log_negssexr | Log of starchy staple food expenditure ratio |
| | hdds_9 | Household dietary diversity score as WFP food grouping |
| | hdds_12 | Household dietary diversity score as FANTA food grouping |
| | hdds_16 | Household dietary diversity score as FAO food grouping |
| | FIES | Food Insecurity Experience Scale |

Source: Authors' own elaboration.

**Table A4.** Indicators for key determinants and dimensions of the resilience of food and nutrition security in Honduras.

| Honduras | | |
|---|---|---|
| **Key Determinant** | **Variable Code** | **Description** |
| ABS | d_ceiling | Acceptable ceiling type |
| | d_walls | Acceptable wall type |
| | d_floor | Acceptable floor type |
| | hh_characteristics_index | Household characteristics Index |
| | d_trash | Acceptable solid disposal |
| | d_water_pri | Acceptable primary water source |
| | d_electricity | Acceptable electricity |
| | d_sewer_sys | Acceptable sewer system |
| | hh_servbasic_index | Basic services index |
| | d_hhtype | Household type |
| | d_kitchen | Kitchen location |
| | d_overcrowding | The acceptable number of people per room |
| | health_service_index | Health service index |

**Table A4.** *Cont.*

| Honduras | | |
|---|---|---|
| **Key Determinant** | **Variable Code** | **Description** |
| AST | d_refrigerador | Refrigerator presence |
| | d_estufa | Stove presence |
| | d_lavadora | washing-machine presence |
| | basic_wealth_index | Primary wealth index (principal value) |
| | d_plancha | Iron presence |
| | d_licuadora | Blender presence |
| | d_tostadora | Toaster presence |
| | higcost_wealth_index | High wealth index (minor value) |
| | d_radio_grabadora | Radiographer presence |
| | d_horno_micro | Microwave presence |
| | d_vhs_dvd | VHS presence |
| | extra_wealth_index2 | Extra wealth index |
| | d_vehiculo | Automobile presence |
| | d_bicicleta | Bicycle presence |
| | d_motocicleta | Moto presence |
| | transport_index | Transport index |
| SSN | t_trans | Income from Cash Remittances |
| | t_famhelp | Income from current aid |
| | t_benefit | Income from Employment Benefits |
| | trans_gov_index | Government transfers Index |
| | t_subsidy | Subsidy income |
| | t_retirement | Retirement income |
| | t_pension | Pension income |
| | subsidy_index | Subsidy index |
| | t_lease | Income from lease |
| | t_lottery | Income from lottery |
| | t_bonos | Household income by vulnerable group bonds |
| | trans_extras_index | Extra transfers index |
| AC | edhd | Education of the household head |
| | edhigh | A higher level of education of the household head |
| | healthhd | IHSS membership of the head of household |
| | hh_head_index | Household Head Index |
| | d_income_rrhh | Human resource or wealth-income |
| | d_other_reme | Income from remittance |
| | d_income_others | Other income sources |
| | participation_index | Participation index |
| | economic_active | Dependency ratio |
| | remesas | Average income from remittances |
| | salary | The average income for salary work |
| | ytraophg | The average income for the primary occupation salary |
| | otras | Average other miscellaneous income |
| FS | log_fexppd | Log of Expenditure per person per day |
| | neg_ssexr | The unfavorable ratio of expenditure on starches to total food |
| | hdds_9 | Household Dietary Diversity Index as PMA food grouping |
| | hdds_12 | Household Dietary Diversity Index as FANTA food grouping |
| | hdds_16 | Household Dietary Diversity Index as FAO food grouping |

Source: Authors' own elaboration.

**Table A5.** Indicators for key determinants and dimensions of the resilience of food and nutrition security in the Dominican Republic.

| Dominican Republic | | |
|---|---|---|
| **Key Determinant** | **Variable Code** | **Description** |
| ABS | d_floor | Acceptable floor type |
| | d_wallsext | Acceptable external wall type |
| | d_kitchen | Acceptable kitchen |
| | hh_characteristics_index | Household characteristics index |
| | d_electricty | Acceptable electricity |
| | d_water | Acceptable water-supply type |
| | d_trash | Acceptable solid disposal |
| | basic_service_index | Basic services index |
| | d_latrine | Acceptable toilet type |
| | d_overcrowding | The acceptable number of people per room |
| | cooking | Energy type for cooking |
| | health_house_index | Health house index |
| AST | d_car | Automobile presence |
| | d_personalcomputer | Computer presence |
| | d_colortelevision | Tv presence |
| | d_refrigerator | Refrigerator presence |
| | d_washingmachine | Washing machine presence |
| | d_inversor | Invertor presence |
| | riquezahogarextra_index | Extra wealth index (principal value) |
| | d_soundequipmet | Sound equipment presence |
| | d_sewingmachine | Sewing machine |
| | d_gasstove | Gas stove presence |
| | d_microwave | Microwave presence |
| | d_dvd | DVD presence |
| | d_fan | Fan presence |
| | riquezahogar_index | House wealth index (minor value) |
| | d_knife | Knife presence |
| | d_peak | Peak presence |
| | d_hoe | Hoe presence |
| | d_bigknife | Big knife presence |
| | tecnologia_index | Technology index |
| | cost house | Monthly dwelling rent |
| AC | educacionpromedio | Average years of education of members of the household |
| | economic_active | The inverse of the dependency ratio |
| | asalariado1 | Primary wage earner's income |
| | asalariado2 | Secondary wage earner's income |
| | independiente1 | Income from the leading independent work |
| | independiente2 | Income from secondary independent work |
| | neoliberal | Income from interest, dividend, and rent |
| | nolaboral2 | Non-work income gambling, inheritance, insurance, etc. |
| | ingresos_dep_index | Income index |

**Table A5.** *Cont.*

| Dominican Republic | | |
|---|---|---|
| **Key Determinant** | **Variable Code** | **Description** |
| SSN | t_comer | Eating comes first |
| | t_alimento | Subsidized food from INSPIRE |
| | t_utiles | School equipment |
| | t_botica | Community pharmacy |
| | t_apagon | Power Outage Reduction Program |
| | t_bombillo | Supply of energy-saving light bulbs |
| | t_gas | Gas subsidy |
| | gobtransfer_index | Government transfers index |
| | remesalocal | Transfers from nation |
| | remesaexterior | Transfers from an external source |
| FS | log_negrazonengel2 | Log of the negative of the Engels ratio |
| | log_negssexr2 | Log of the negative of Bennett's ratio |

Source: Authors' own elaboration.

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
