# Peer review of "Household Resilience to Food and Nutrition Insecurity in Central America and the Caribbean"

_sustainability, doi:10.3390/su13169086_

Round 1

Reviewer 1 Report

The authors assess the resilience to food and nutrition insecurity of households in five Latin American and Caribbean countries, placing particular emphasis on farm diversification as a means of improving diet diversification and resilience. The results show both similar constraints and divergences between and within countries and the authors conclude that these results can be useful to improve resilience-enhancing interventions.

The article is relevant and timely. The paper is straightforward, well-written and well structured. I find particularly interesting the discussion of results and conclusions even if at some points they are not fully supported by the results. Notwithstanding this, I think they are rich in very interesting and useful insights. Methods are appropriate and adequately described. I only miss a clearer (and short) explanation on whether the corresponding FNSI measures food quantity or quality. Not making this clear may lead to confusion at certain points when reading the paper. Figures and tables are complete, informative and clearly presented.

I include below a couple of additional and minor comments which, if addressed, could improve the paper:

  • Line 97: The authors state “2006-2007” in the body of the text but “2005-2006” in Table 1.
  • Line 110: The authors state “98.4 percent” in the body of the text but “98.1%” in Table 1.
  • Line 114: There is no income included for Guatemala in Table 1 (questionnaire section).
  • Lines 115-116: ICTs are included in Table 1 for El Salvador, not for Costa Rica.
  • Line 165: ?ast has been listed twice, while no ?ac has been included.
  • Lines 217-228: I would suggest explaining this part a bit more clearly.
  • Lines 293-294: and except for Honduras.
  • Lines 300-302: Only four of the five reference territories have been listed (Guatemala´s is missing, I think).
  • Lines 303-304: I cannot see in Table 3 “for Guatemala and Honduras, the resilience difference for households by the mother tongue of household head”.
  • Line 325: What does “from the part of Guatemala” mean?
  • Line 327: The word “similarly” leads to confusion.
  • Line 336-338: I think the sentence is not entirely true.
  • Line 341: With more secondary school-aged children too?
  • Line 344: According to Table 3, it should be Guatemala instead of Honduras.
  • Line 347: Data are not available for Honduras either.
  • Lines 349-350: At points, the difference between determinants and indicators is not clear (as in this sentence. Do the authors mean determinants or indicators here?).
  • Line 356: There is no “teaching of household heads in Costa Rica” included in Figure 2 (in the A.C. graph).
  • Lines 377-378: There is evidence of a negative resilience level for rural areas only for three countries.
  • Line 396: Does Table 2 really show a negative relation between resilience and dietary diversity?
  • Lines 506-510: I would suggest rewriting this part in a clearer way. It is not clear which country each statement refers to.
  • Tables A1, A2, A3, A4 and A5: What do RPS and RYAN stand for? In addition, AST is written as ACT in Tables A1, A2, A3, A4 and A5 and ABS as ASB in Tables A2, A3, A4 and A5. I would suggest standardizing these terms throughout the paper.

Author Response

Dear reviewers, thank you for taking the time to read and comment on our paper. We value your inputs and work and think that the paper reads much better now. We have accepted all the requests made by Reviewer 1 and Reviewer 2. Except few for which we provide response here.

R1

  • Line 114: There is no income included for Guatemala in Table 1 (questionnaire section).
    • We refer to income-generating activities not income per se
  • Line 341: With more secondary school-aged children too?
    • These are weird findings, I agree, but this is what data are telling us

Reviewer 2 Report

The work reports Household resilience to food and nutrition insecurity in Central America and the Caribbean. The work improves and fills the knowledge gap and informs on the application of the Resilience Index Measurement and Analysis version II (RIMA-II) for estimating Resilience on Food and Nutrition Security Indicators (FNSI) in five vulnerable countries in Central America and the Caribbean: Costa Rica, El Salvador, Guatemala, Honduras, and the Dominican Republic.

The work is interesting and methodologically well set up, but it is difficult to read and needs a thorough revision of English. It is difficult to read because it also requires changes in literature citation in the whole text. Please, see Instruction for Authors, and cite the text adequately.

Some additional suggestions:

line 56 – try to avoid the phrase “to the best of our knowledge”

line 67 – try to additionally describe what are you mean in this part “highly heterogeneous in agriculture and vulnerability…” Many people do not have an idea of this part of Caribbean countries.

line 69-70 – please try to implicate this scope in the abstract in a good manner

line 88-90  - please delete

Table 1 – row Samples divide in two, one for number, one for percents

line 105 and other – replace „percent“ to „%“

Table 3 and other – please insert a note for „***“ as for Table 1

Try to uniform each graph with an MDPI template/font, as well as improve the quality of all figures.

Author Response

Dear reviewers, thank you for taking the time to read and comment on our paper. We value your inputs and work and think that the paper reads much better now. We have accepted all the requests made by Reviewer 1 and Reviewer 2. Except few for which we provide response here.

  • line 67 – try to additionally describe what are you mean in this part “highly heterogeneous in agriculture and vulnerability…” Many people do not have an idea of this part of Caribbean countries.
    • We think that the reference provided can explain the heterogeneity adequately. I am not sure that providing more background here would make the reading of this (already overly complex) paper easier.
  • line 69-70 – please try to implicate this scope in the abstract in a good manner
    • I think the last sentence of the abstract presents this research question. However, I am happy to modify it if needed.